# PgRsp Is a Novel Redox-Sensing Transcription Regulator Essential for *Porphyromonas gingivalis* Virulence

**DOI:** 10.3390/microorganisms7120623

**Published:** 2019-11-28

**Authors:** Michał Śmiga, Teresa Olczak

**Affiliations:** Laboratory of Medical Biology, Faculty of Biotechnology, University of Wroclaw, F. Joliot-Curie 14A St., 50-383 Wroclaw, Poland

**Keywords:** *Porphyromonas gingivalis*, Crp/Fnr, CooA, PgRsp, heme, redox, transcription regulation, virulence

## Abstract

*Porphyromonas gingivalis* is one of the etiological agents of chronic periodontitis. Both heme and oxidative stress impact expression of genes responsible for its survival and virulence. Previously we showed that *P. gingivalis* ferric uptake regulator homolog affects expression of a gene encoding a putative Crp/Fnr superfamily member, termed *P. gingivalis* redox-sensing protein (PgRsp). Although PgRsp binds heme and shows the highest similarity to proteins assigned to the CooA family, it could be a member of a novel, separate family of proteins with unknown function. Expression of the *pgrsp* gene is autoregulated and iron/heme dependent. Genes encoding proteins engaged in the oxidative stress response were upregulated in the *pgrsp* mutant (TO11) strain compared with the wild-type strain. The TO11 strain showed higher biomass production, biofilm formation, and coaggregation ability with *Tannerella forsythia* and *Prevotella intermedia*. We suggest that PgRsp may regulate production of virulence factors, proteases, Hmu heme acquisition system, and FimA protein. Moreover, we observed growth retardation of the TO11 strain under oxidative conditions and decreased survival ability of the mutant cells inside macrophages. We conclude that PgRsp protein may play a role in the oxidative stress response using heme as a ligand for sensing changes in redox status, thus regulating the alternative pathway of the oxidative stress response alongside OxyR.

## 1. Introduction

Periodontal diseases, a group of the most abundant human infectious diseases [1,2,3,4], are characterized by reabsorption of tooth-supporting tissues caused by gingiva inflammation induced by virulence factors produced by periodontopathogens and the exaggerated response of the host immune system. There is also growing evidence supporting a link between periodontal diseases and several systemic diseases, such as cardiovascular and respiratory diseases, rheumatoid arthritis, diabetes, osteoporosis, and Alzheimer′s disease [5,6,7,8,9,10]. *Porphyromonas gingivalis*, one of the agents involved in initiation and progression of chronic periodontitis [2,3,4,11], is a Gram-negative, obligatory anaerobic, asaccharolytic bacterium, residing in periodontal pockets of the human oral cavity.

*P. gingivalis* has developed numerous sophisticated mechanisms for the acquisition of nutrients, enabling its survival as well as its proliferation inside and spread between host cells [12,13,14,15,16]. Among the most important virulence factors are proteases, mainly lysine- and arginine-specific gingipains encoded by *kgp*, *rgpA*, and *rgpB* genes, which play a crucial role in the degradation of host proteins and tissue components [17,18,19]. Together with the heme-uptake Hmu system, gingipains engage in heme acquisition, which is essential for *P. gingivalis* survival due to the lack of ability to synthesize protoporphyrin IX [12]. Other virulence factors produced by *P. gingivalis* are hemagglutinins and fimbriae, which are responsible for interaction with host cells, as well as with other bacteria, enabling efficient *P. gingivalis* colonization of the oral cavity [18,19,20,21,22].

Bacteria residing inside the human oral cavity are exposed to environmental stresses. Among them are oxygen, reactive oxygen species (ROS), and reactive nitrogen species (RNS), which can be detrimental, especially to anaerobic bacteria. They damage proteins, lipids, and nucleic acids, impairing their function and leading to bacterial death [23,24]. Moreover, specialized human cells, such as granulocytes or macrophages, produce antimicrobial particles, which include O_2_^–^, H_2_O_2_, NO_2_, ONOO^−^, and N_2_O_3_ [25,26,27]. Therefore, bacteria have developed systems that sense redox conditions of the extracellular environment and lead to changes in the expression of genes responsible for protection against these harmful agents. The most abundant bacterial transcription factor responsible for the oxidative stress response is OxyR protein [23,28,29]. *P. gingivalis* possesses an OxyR homolog which is important for both H_2_O_2_ resistance and aerotolerance [30].

Recently, we have shown that *P. gingivalis* ferric uptake regulator homolog (PgFur) is an important transcription factor, regulating genes involved in interaction with host cells and other periodontopathogens [13,16]. A *P. gingivalis* mutant strain lacking a *pgfur* gene showed reduced tolerance to H_2_O_2_ and was more sensitive to air exposure, although we did not observe regulation of typical genes responsible for the oxidative stress response [13,16]. We discovered that the expression of many genes involved in regulation of gene expression was altered in the mutant strain, including the downregulated *PGA7_00004090* gene, encoding a putative transcription factor belonging to the cAMP receptor protein/fumarate and nitrate reductase regulator (Crp/Fnr) superfamily. Proteins from the Crp/Fnr superfamily belong to global transcription regulators involved in the regulation of genes responsible for cellular metabolism and the response to environmental stresses, such as oxidative or nitrosative stress [31]. Some of them bind sensor molecules, such as CO or NO, using heme complexed to these proteins [32,33]. Proteins from the Crp/Fnr superfamily are characterized by a similar structure, composed of a C-terminal domain with a helix-turn-helix motif engaged in DNA binding, and an N-terminal domain involved in ligand binding. Both domains are linked by an alpha-helix region responsible for protein dimerization [32,34]. Despite structural similarity, they demonstrate different functions with a variety of activation mechanisms.

Precise sensing of redox state is crucial for human oral anaerobic pathogens in proceeding with the virulence processes. It is likely that *P. gingivalis* may utilize other redox-sensing systems along with the classical OxyR-dependent system. Therefore, in this study we characterized a *P. gingivalis* protein encoded by the *PGA7_00004090* gene and investigated its involvement in virulence. We found that it is a member of a novel family of Crp/Fnr superfamily transcription regulators which regulates production of *P. gingivalis* virulence factors, such as proteases, Hmu heme acquisition system, and FimA protein. Growth retardation of the mutant strain under oxidative conditions and decreased survival ability inside macrophages suggests that this protein is a heme-based redox sensor which plays a role in the oxidative stress response and *P. gingivalis* virulence.

## 2. Materials and Methods

### 2.1. Bacterial Strains and Growth Conditions

*P. gingivalis* wild-type (A7436) [35], *pgfur* mutant (TO6) [13], *pgrsp* mutant (TO11), and complemented *pgrsp* mutant (TO11 + *pgrsp*) strains were grown anaerobically at 37 °C for 5 days on blood agar plates (Schaedler broth with hemin and cysteine, supplemented with 5% sheep blood and menadione; Biomaxima, Lublin, Poland) as described previously [36]. These cultures were used as the inoculum for growth in liquid basal medium, composed of 3% trypticase soy broth (Becton Dickinson, Heidelberg, Germany), 0.5% yeast extract (Biomaxima), supplemented with 0.05% cysteine and 0.5 µg/mL menadione (BM). To imitate low-iron/heme conditions of the healthy oral cavity and initial periodontitis stage, hemin was not added and 160 µM 2.2-dipyridyl was used as an iron chelator (BM + DIP). To ensure high-iron/heme conditions of the oral cavity affected by advanced periodontitis, 7.7 µM hemin chloride was added (BM + Hm). Mutant strains were maintained in the presence of 1 µg/mL erythromycin and complemented mutant strains in the presence of 1 µg/mL of tetracycline and 1 µg/mL erythromycin. *Prevotella intermedia* strain 17 (GenBank: CP003502.1 and CP003503.1) and *Tannerella forsythia* (ATCC 43037) were maintained as described previously [37]. *Streptococcus gordonii* (ATCC 10558) was grown in TSB agar or liquid medium (5% TBS) under an increased concentration of CO_2_ using the Atmosphere generation system (Thermo Scientific, Waltham, MA, USA) as reported previously [16].

### 2.2. Generation of Mutant and Complemented Mutant Strains

The *pgrsp* (*PGA7_00004090*) mutant (TO11) strain was constructed in the *P. gingivalis* A7436 strain by replacement of the majority of the gene by an erythromycin-resistance cassette (*ermF*) from *Bacteroides fragilis* as described previously [36]. Upstream and downstream flanking fragments of the *pgrsp* gene were PCR amplified using genomic DNA isolated from the *P. gingivalis* A7436 strain, and the *ermF* gene was PCR amplified using pTIO-1 plasmid [38] and primers listed in Appendix A. Obtained sequences were ligated using NEBuilder HiFi DNA Assembly (New England Biolabs, Ipswich, MA, USA). Linear DNA was used for electroporation of the wild-type A7436 strain [39]. To generate the complemented mutant strain (TO11 + *pgrsp*), the *pgrsp* gene together with the promoter region was amplified using PCR and subsequently cloned into XhoI and BamHI restriction sites of a pTIO-tetQ plasmid [40] and introduced into the TO11 mutant strain by electroporation. Homologous recombination was confirmed by PCR and DNA sequencing (Microsynth, Balgach, Switzerland).

### 2.3. Oxidative Stress Formation

Overnight cultures of *P. gingivalis* grown in BM + Hm medium were used to inoculate basal medium supplemented with 0.5 µg/mL menadione and 7.7 µM hemin chloride, with (BM + Cys + Hm) or without (BM–Cys + Hm) cysteine, at starting optical density at 600 nm (OD_600_) ~0.2. Bacteria were grown with or without addition of 0.05 mM or 0.25 mM H_2_O_2_ added at the culture starting point. Alternatively, H_2_O_2_ was added into 4-h bacterial cultures.

### 2.4. Gene Expression Analysis

Gene expression was initially examined in bacteria cultured for 20 h in iron/heme rich (Hm) or depleted (DIP) medium using microarray analysis as reported previously [13,16]. Additionally, reverse transcriptase–quantitative polymerase chain reaction (RT-qPCR) was used as reported previously [16,40]. Total RNA was isolated from 5 × 10^7^ to 4 × 10^8^
*P. gingivalis* cells grown for 4, 10, and 24 h in iron/heme rich or iron/heme depleted medium, or one hour after exposure to oxidative stress using the Total RNA Mini Kit (A&A Biotechnology, Gdynia, Poland) and the Clean-Up RNA Concentrator Kit (A&A Biotechnology). Reverse transcription was carried out using total RNA and random hexamers with the SensiFAST cDNA Synthesis Kit (Bioline, London, UK). qPCR was performed using the SensiFAST SYBR No-ROX Kit (Bioline) and LightCycler 96 (Roche, Basel, Switzerland). All primers are listed in Appendix A. Samples were analyzed in triplicate in two biological replicates and melting curves were generated to measure the quality of amplified products. Relative changes in gene expression were determined using LightCycler 96 software and *P. gingivalis 16S rRNA* (*PGA7_00000960*) as the reference gene.

### 2.5. Homotypic Biofilm Formation and Coaggregation Assay

*P. gingivalis* was grown in 96-well plates as described previously [41]. Briefly, fresh BM + Hm or BM + DIP medium was inoculated with the overnight cultures at starting OD_600_ ~0.2 and bacteria were grown for 24 h. Non-attached bacterial cells were washed out three times with 20 mM phosphate buffer, pH 7.4, containing 140 mM NaCl (PBS). Then, 1% crystal violet (Roth, Frederikssund, Denmark) was used to visualize the biofilm formed.

Coaggregation assay was performed as described previously [16]. Briefly, overnight liquid cultures of *P. gingivalis*, *P. intermedia*, *T. forsythia,* and *S. gordonii* were centrifuged (8000× *g*, 20 min, 4 °C) and washed with PBS. Subsequently, optical density at 600 nm (OD_600_) of the cultures was adjusted to 1.0 using PBS. The cultures of *P. gingivalis* and respective bacteria were mixed at a 1:1 ratio and incubated anaerobically for 6 h. The coaggregation rate was monitored by measurement of the decrease in OD_600_. Autoaggregation of the monocultures was used as a control.

### 2.6. Proteolytic Activity Assay

To determine total proteolytic activity of *P. gingivalis* cultures, azocasein (Sigma-Aldrich, St. Louis, MO, USA) as the proteolytic substrate was used as described previously [16]. Briefly, 50 μL of 2.5% azocasein solution and 30 μL of 0.5% NaHCO_3_ solution (pH 8.3) were added to 20 μL of *P. gingivalis* culture samples, following by incubation at 37 °C for 30 min. The reaction was stopped by precipitation of undigested proteins with 400 μL of 5% trichloroacetic acid and then centrifuged (10,000× *g*, 5 min). Subsequently, 240 μL of 0.5 M NaOH was added to 400 μL of the supernatant and absorbance at 440 nm was measured. Proteolytic activity exhibited by the wild-type strain under given conditions was considered as 100%.

### 2.7. Cell Culture and Infection Assay

THP-1 (human acute monocytic leukemia) (ATCC TIB-202) cell line was obtained from the American Type Culture Collection (ATCC). THP-1cells were grown in 12-well plates in RPMI-1640 medium (BioWest, Nuaillé, France) with the addition of 2 mM L-glutamine (BioWest), 10% heat-inactivated fetal bovine serum (BioWest), 100 U/mL penicillin and 100 µg/mL streptomycin as described previously [13]. THP-1 cells were differentiated toward macrophages by supplementing the culture medium without antibiotics with 0.01 µg/mL phorbol 12-myristate 13-acetate (PMA) (Sigma-Aldrich) for 48 h prior to infection.

*P. gingivalis* cells were grown to the early stationary phase in BM + Hm medium, centrifuged (4000× *g*, 20 min, 4 °C) and washed twice with PBS. Macrophages were infected with *P. gingivalis* at multiplicity of infection (MOI) of 100 [16]. After 4 h, the cells were washed twice with PBS and one out of the three portions was lysed with water to count live bacterial cells present inside macrophages and attached to them. The two remaining portions were further incubated with the fresh culture medium, supplemented with 300 µg/mL gentamicin and 200 µg/mL metronidazole. After 1 h, one portion of the cells was washed three times with PBS and lysed with water to count live bacterial cells residing inside macrophages. The last portion of the cells was collected 8 h after infection, washed three times with PBS, and lysed with water to count live bacterial cells present inside macrophages. All samples were serially diluted and plated onto Schaedler blood agar plates to determine colony-forming units (CFU).

### 2.8. Plasmid Construction, Overexpression, and Purification of Recombinant PgRsp Protein

For overexpression of PgRsp protein in fusion with N-terminally attached His-tag and maltose-binding protein (MBP), pMAL_c5x plasmid (New England Biolabs), modified as described previously (pMAL c5x_His plasmid), was used [40]. The *pgrsp* gene was amplified by PCR using primers listed in Appendix A and cloned into the pMAL c5x_His plasmid using restriction-free cloning [42]. Final expression pMAL c5x_PgRsp plasmids were verified by DNA sequencing (Microsynth).

To obtain modified PgRsp protein with single amino acid substitutions, appropriate modifications of pMAL c5x_PgRsp plasmid were performed using the QuikChange II Site-Directed Mutagenesis Kit (Agilent Technologies, Santa Clara, CA, USA) and primers listed in Appendix A. The resulting plasmids were sequenced and used to transform *E. coli* cells.

PgRsp protein variants in fusion with 6His-MBP protein were overexpressed in *E. coli* Rosetta 2 (DE3) strain grown in Terrific Broth (TB), supplemented with 35 µg/mL chloramphenicol and 100 µg/mL ampicillin. After transformation, bacteria were grown at 37 °C with shaking at 220 rpm until cultures reached OD_600_ = 1.0. Subsequently, cultures were cooled down to 16 °C and protein overexpression was induced by addition of IPTG (Roth) to a final concentration of 0.5 mM. Proteins were overexpressed for 16–20 h. The cells were collected by centrifugation (4000× *g*, 20 min, 4 °C) and the pellets were stored at −20 °C until used. The bacteria were re-suspended in 25 mM HEPES, pH 7.8, containing 300 mM NaCl, and lysed by sonication (UP100H Hielscher Ultrasonics, at 4 °C). Then, the lysate was centrifuged (40,000× *g*, 15 min, 4 °C) and the soluble fraction was loaded onto an amylose resin (New England Biolabs) column. After washing, 6His-MBP-PgRsp protein was eluted with 25 mM HEPES, pH 7.8, containing 300 mM NaCl and 25 mM maltose. The protein was then transferred to 25 mM HEPES, pH 7.8, containing 80 mM NaCl, 5% glycerol and 2 mM CaCl_2_ using Amicon Ultra-15, 10.000 NMWL (Millipore, Burlington, MA, USA). Subsequently, the protein was subjected to digestion with Factor Xa (New England Biolabs) according to the manufacturer’s protocol (New England Biolabs), but with addition of 0.01% SDS. Untagged PgRsp protein was obtained by binding of released 6His-MBP to TALON resin (Clontech, Mountain View, CA, USA). Purified PgRsp protein was stored in 25 mM HEPES, pH 7.8, containing 300 mM NaCl and 10% glycerol at −20 °C until used. Protein purification process was monitored by sodium dodecyl sulfate–polyacrylamide gel electrophoresis (SDS-PAGE) in 12% polyacrylamide gels and staining with Coomassie Brilliant Blue G-250 (CBB G-250). Concentration of PgRsp protein was determined by measuring absorbance at 280 nm using the empirical molar absorption coefficient (ε_280_ = 15.33 mM^−1^ cm^−1^) determined in this study as reported by others [43].

### 2.9. Protein–Heme Complex Formation

Hemin chloride (ICN Biomedicals, Costa Mesa, CA, USA) solution was freshly prepared and protein–heme complexes were formed as described previously [20]. UV-visible spectra were monitored using a double beam Jasco V-750 spectrophotometer (10 mm path length). Heme binding to the PgRsp protein (5 or 10 µM in PBS) was analyzed under air (oxidized) or reduced conditions, the latter formed by addition of 10 mM sodium dithionite (Sigma-Aldrich) and a layer of mineral oil (Sigma-Aldrich).

### 2.10. Electrophoretic Mobility Shift Assay

For the electrophoretic mobility shift assay (EMSA), the LightShift Chemiluminescent EMSA Kit (Thermo Scientific) was used according to the manufacturer′s protocol. Briefly, biotin-labeled DNA fragments of selected promoter regions were amplified by PCR using primers listed in Appendix A. Then, 1 ng of DNA was added to 1× binding buffer with the addition of 2.5% glycerol, 5 mM MgCl_2_, 50 ng/μL poly (dI-dC) and varying concentrations of heme, purified PgRsp protein, non-biotinylated DNA and sodium dithionite. Samples were incubated for 20 min at room temperature, and subsequently subjected to electrophoresis for 1 h at 150 V in 0.25× TBE buffer (25 mM Tris, 25 mM boric acid, 0.5 mM EDTA, pH 8.6) on pre-run for 30 min 6% polyacrylamide gels. After electrophoresis, samples were transferred onto a nylon membrane (Bionovo, Legnica, Poland) using wet transfer in the presence of 1× TBE buffer (300 mA, 30 min, 4 °C) and subjected to crosslinking for 10 min with UV radiation at a wavelength of 274 nm. The protocol was continued according to the manufacturer′s (Thermo Scientific) guidelines and labeled nucleic acids were visualized using the appropriate reagents and Chemidoc MP Imaging System (Bio-Rad, Hercules, CA, USA).

### 2.11. Phylogenetic Analyses

To ascribe the protein encoded by the *PGA7_00004090* gene to a family of the Crp/Fnr superfamily, phylogenetic analysis was performed using comparison of homologous sequences defined by BLAST search (NCBI server). The sequences were aligned by Clustal Omega and the phylogenetic tree was constructed using the EvolView server [44,45,46]. The prediction of PgRsp protein structure was performed using the Phyre2 server [47] and the Swiss PDB Viewer program [48] was used to construct three-dimensional protein models.

### 2.12. Statistical Analysis

Unpaired Student′s *t*-test or one-way ANOVA (analysis of variance) with post hoc Tukey honest significant difference comparison was used to analyze the data (GraphPad Prism version 5). Results are shown as mean ± standard deviation. All experiments were performed at least three times in two replicates. Statistically significant results were indicated with asterisks (* *p* < 0.05; ** *p* < 0.01; *** *p* < 0.001).

## 3. Results

### 3.1. PgRsp—a Heme-Binding Member of a Novel Family of Crp/Fnr Superfamily Transcription Regulators

To assign the potential function of the PgRsp protein, first bioinformatics approaches were employed. Searching for PgRsp-encoding sequences in *P. gingivalis* strains revealed that amino acid sequences deposited in databases for several *P. gingivalis* strains (e.g., W83, AJW4, ATCC 33277, 381, HG66, A7A1-28) predicted a 207-amino-acid version of the PgRsp protein, whereas in the A7436 strain a 283-amino-acid protein was predicted [49]. Our theoretical approaches and preliminary analysis of the 5′UTR region of the mRNA encoding PgRsp using 5′RACE (data not shown) led us to the conclusion that in the A7436 strain the *pgrsp* gene encodes a 207-amino-acid long protein. Therefore, in this study for further theoretical and experimental approaches we used this protein version.

Next, the amino acid sequence of the PgRsp protein was compared with homologous proteins from different families of the Crp/Fnr superfamily. Construction of the phylogenetic tree showed that the protein encoded by the *pgrsp* gene was not ascribed to any of the best-characterized families. The highest amino acid sequence similarity (~5–20%) was found when compared with proteins belonging to the CooA and NnrR families (Figure 1 and Figure 2). Alignment of the sequences of the closest homologs to the PgRsp protein showed 19.5% and 15.1% homology to proteins from *Rhodobacter sphaeroides* and *Brucella canis*, respectively, assigned as members of the NnrR family, as well as 20.5% and 20% homology to the best characterized CooA family proteins from *Rhodospirillum rubrum* (RrCooA) and *Azotobacter vinelandii*, respectively.

Some proteins belonging to the Crp/Fnr superfamily, including CooA and Dnr proteins, may bind heme [31,33,50]. Preliminary observations based on the visual inspection of the color exhibited by the purified PgRsp protein samples confirmed, at least in part, such a possibility. To ascertain whether PgRsp is a heme-binding protein, the purified protein was titrated with heme under air (oxidized) or reduced conditions. The λ_max_ in the Soret region for the PgRsp–heme complex under oxidized or reduced conditions was 412 or 425 nm, respectively (Figure 3a,b). The maxima for the Q bands were clearly seen at 531 and 559 nm only for the PgRsp–heme complex under reduced conditions (Figure 3b). In contrast, under oxidized conditions weak maxima at 533 and 620 nm were observed (Figure 3a,b). Similar spectra were reported for RrCooA protein [51]. To further study the PgRsp–heme complex, the difference spectra were generated (Figure 3c,d). Although maxima of Q bands under oxidized conditions were present at 537 and 563 nm and an additional maximum at 648 nm (Figure 3c), reduction of the heme significantly increased Q band maxima at 529 and 560 nm (Figure 3d). However, no changes in UV-visible spectra were observed when Cys76, Met78, or His85 was singly replaced by Ala residue (data not shown). In addition, all the purified PgRsp protein variants were visible as red colored samples, suggesting the presence of heme bound complexes (data not shown).

Further, the PgRsp amino acid sequence was compared with sequences of representative homologous proteins from the CooA family, as they showed the highest amino acid similarity (Figure 1 and Figure 2). The heme-binding motifs present in CooA proteins include Cys or His and Pro or Ala residues (Figure 2). Although amino acid sequence comparison showed the possibility of a similar motif for heme binding in the PgRsp, the structure modeling based on RrCooA protein did not include it. Therefore, a theoretical three-dimensional model of PgRsp was constructed, using known structures of proteins from the Crp/Fnr superfamily as templates (Figure 4) and for final modeling, Crp homolog from *Eubacterium rectale*, showing the highest amino acid sequence coverage and homology (Figure 2) was used (PDB ID: 3DV8). The modeled structure showed fold typical for Crp proteins, comprising N- and C-terminal (DNA-binding domain) domains, both connected by an alpha helix (Figure 4). Besides Cys and His, CooA homologs use N-terminal Pro or Ala as the second heme-binding ligand in the non-active state [52,53]. Neither PgRsp nor its closest homologs have such an amino acid residue at the N terminus (Figure 2).

Our results suggest that PgRsp protein may bind heme in a different manner and thus exhibit a different mechanism of gene regulation at the molecular level compared to the known heme-binding proteins comprising the Crp/Fnr superfamily.

### 3.2. Pgrsp Gene Expression is Iron/Heme- and PgFur-Dependent

The *pgrsp* gene expression analysis carried out in the wild-type A7436 strain in relation to iron/heme availability and bacterial growth phase demonstrated that it was significantly increased in bacteria cultured for a prolonged time in the iron/heme-depleted medium (1.76-fold change determined by microarray analysis and data shown in Figure 5a). Analysis of the influence of growth phase on *pgrsp* gene expression showed the highest levels of mRNA encoding PgRsp produced in bacteria cultured under heme/iron excess during the middle exponential growth phase (10 h) in comparison to the early growth phase (4 h), whereas the lowest mRNA levels were produced at the early stationary phase (24 h) (Figure 5b). No significant influence of growth phases was observed on the *pgrsp* gene expression when bacteria were grown under low-iron/heme conditions.

Recently, we reported that *P. gingivalis* PgFur is essential for *P. gingivalis* virulence and regulates, among many other functions, expression of the *pgrsp* gene [13,16]. Therefore, we suggest that PgRsp may cooperate with PgFur protein in regulation of bacterial virulence. This suggestion was supported by microarray analysis demonstrating decreased *pgrsp* gene expression in the *pgfur* mutant strain, especially when bacteria were grown in iron/heme-depleted conditions (−2.78-fold change).

### 3.3. PgRsp Plays a Role in Regulation of P. gingivalis Virulence

To characterize the importance of the PgRsp protein in the production of *P. gingivalis* virulence factors, a *pgrsp* deletion mutant strain (TO11) and complemented mutant strain (TO11 + *pgrsp*) were constructed, and expression of the virulence factors chosen in regard to their importance in the colonization of the oral cavity and participation in synergistic mechanisms providing nutrients for *P. gingivalis* and other periodontopathogens was analyzed. Examination of bacterial growth rates showed slight differences between TO11 and A7436 strains grown in iron/heme-rich medium. The mutant TO11 strain grew similarly to the wild-type A7436 strain until it reached the early stationary phase, exhibiting higher biomass production during prolonged growth time (Figure 6a). Complementation of the TO11 strain (Figure 6b) resulted in partial correction of the mutant’s phenotype (Figure 6a).

In order to initiate infection, *P. gingivalis* must interact with other periodontopathogens and host cells. Important virulence factors which play a role in this process are hemagglutinins, gingipains, and fimbriae [17,19,21,54]. To examine the potential influence of PgRsp on the production of *P. gingivalis* surface proteins, coaggregation ability of *P. gingivalis* and selected oral bacteria, *P. intermedia*, *T. forsythia*, and *S. gordonii,* was analyzed. We observed faster coaggregation of the mutant TO11 strain with *P. intermedia* and *T. forsythia* in comparison to the wild-type A7436 strain (Figure 6c). No differences were observed in interaction of TO11 and A7436 strains with *S. gordonii* (Figure 6c). Additionally, the homotypic biofilm formation ability of both strains was monitored. In comparison to the A7436 strain, the TO11 strain formed higher biofilm structures (Figure 6d), which could be correlated with higher bacterial biomass production in the case of the TO11 strain (Figure 6a).

To determine whether fimbriae could be involved in this process, expression of the *fimA* gene, encoding the main component of *P. gingivalis* long fimbriae [21] was examined. Surprisingly, *fimA* gene expression was lower in the TO11 strain in comparison to the A7436 strain (Table 1). It is likely that lower amounts of fimbriae could enable exposure or induce higher expression of other proteins involved in the interaction between *P. gingivalis* and other bacterial cells, finally increasing the coaggregation capacity of the mutant cells.

Gingipains and Hmu heme-acquisition system proteins are considered the most important virulence factors of *P. gingivalis* [12,17,55,56,57]. Therefore, expression of *hmuY*, *rgpA*, *rgpB* and *kgp* genes in the TO11 strain in comparison to the A7436 strain was examined. Although no significant changes in the *hmuY* gene expression were observed (Table 1), higher expression of the *kgp* gene and lower expression of *rgpB* and *rgpA* genes were found, especially in bacteria grown under low-iron/heme conditions. During prolonged culture in iron/heme-rich medium, expression of *rgpB* and *rgpA* genes was slightly higher in the *pgrsp* mutant strain in comparison to the wild-type strain (Table 1). Moreover, the expression of the *PGA7_00031810* gene encoding a putative protease was examined and it showed decreased expression in the TO11 strain (Table 1). To correlate gene expression with the activity of gene products, total proteolytic activity was determined. The TO11 strain showed higher proteolytic activity in iron/heme-rich medium and lower in iron/heme-depleted medium (Figure 6e).

Altogether, these results suggest that PgRsp might regulate expression of important virulence factors of *P. gingivalis.* Based on our results one may assume that PgRsp is involved in the control of expression of genes engaged in interaction between *P. gingivalis* cells and other periodontopathogens.

### 3.4. PgRsp is Involved in the Response to Oxidative Stress

Sequence similarity of PgRsp to CooA proteins and the observed phenotype of the mutant *pgrsp* strain led us to conclude that PgRsp may be involved in sensing extracellular stresses. Therefore, the expression of selected genes, namely *oxyR*, *sod*, *tpx*, *bcp*, *pgdps*, *ahpC, rbr, pgfur*, and *ustA,* was analyzed. We observed increased expression of all but *pgdps* and *pgfur* genes in the TO11 strain in comparison to the A7436 strain during prolonged growth, especially under iron/heme-depleted conditions (Table 1). To further examine the potential role played by PgRsp particularly in the oxidative stress response, we analyzed the influence of different redox conditions on bacterial growth. Bacteria were cultured in rich culture media but in the absence of cysteine (culture medium component providing reduced conditions and protecting *P. gingivalis* from atmospheric oxygen) and with different H_2_O_2_ concentrations, added at the initiation of bacterial cultures. We observed growth retardation of the TO11 strain compared to the A7436 strain during bacterial growth in rich culture media lacking cysteine (conditions inducing mild aerobic stress) and in rich culture media lacking cysteine and with the addition of 0.05 mM H_2_O_2_ (conditions inducing high oxidative stress). Higher concentration of H_2_O_2_ (0.25 mM) was lethal for the TO11 strain whereas the A7436 strain showed only a slight delay in the growth ability (Figure 7a). Additionally, the effect of oxidative stress generated in a later stage of bacterial cultures was tested. Surprisingly, no effect was observed when bacteria lacking PgRsp protein were grown for 4 h and subsequently treated with H_2_O_2_ (Figure 7b).

Resistance of *P. gingivalis* to oxidative stress is an important factor not only for survival in the oral microbiome but also inside host immune cells. Using THP-1-derived macrophages as model cells, we studied the role of PgRsp protein in invasion of host immune cells. We observed that attachment of the mutant TO11 strain to macrophages did not differ from that observed for the wild-type A7436 strain (Figure 8). However, lower invasion of and survival ability inside macrophages of the *pgrsp* mutant cells were observed, suggesting impaired capability to enter macrophages and/or resist antimicrobial mechanisms inside macrophages.

To further understand the involvement of PgRsp in the regulation of gene expression during exposure to oxidative stress, expression of selected genes under various redox conditions was tested. Surprisingly, the largest changes in gene expression were observed when culturing bacteria in BM medium without cysteine (BM–Cys + Hm) (Table 2), with a significant decrease in expression of genes involved in bacterial resistance to oxidative stress (e.g., *sod*, *bcp*, *tpx*). When bacteria were exposed to 0.25 mM H_2_O_2_, higher expression of *oxyR*, *tpx*, and *rbr* genes and decreased expression of *bcp*, *pgdps*, and *ahpC* genes were observed. When bacteria were cultured in rich culture medium supplemented with cysteine (BM + Cys + Hm), gene expression did not change as dramatically as in bacteria exposed to oxidative stress. This could suggest that PgRsp protein is an important factor responsible for the protection of *P. gingivalis* during exposure to agents disturbing redox homeostasis. Since the phenotype of the bacteria lacking this protein was significantly altered when exposed to oxidative stress, we termed this protein a *P. gingivalis* redox-sensing protein (PgRsp).

Besides the downregulation of genes responsible for protection against oxidative stress, we observed downregulation of *hmuY* and *fimA* genes, and genes encoding gingipains during exposure of bacteria to air, implying regulation of those genes by the PgRsp protein. During exposure of *P. gingivalis* to H_2_O_2_, the mutant TO11 strain showed over 7 times higher expression of the *oxyR* gene. This could suggest that increased production of OxyR protein may compensate the lack of PgRsp, which results in higher expression of genes connected with protection against oxygen, namely *sod*, *rbr*, and *tpx*.

### 3.5. PgRsp Regulates Hmu Operon, Bcp Genes and its own Expression

To confirm whether regulation of selected genes, whose expression was significantly changed in the mutant TO11 strain in comparison to the wild-type A7436 strain was direct, the interaction of the PgRsp protein with fragments of promoters of the *hmu* operon and *bcp* gene using EMSA analysis was analyzed. Retardation in migration of promoter fragments after addition of the PgRsp protein (Figure 9a,b) demonstrated that the protein may regulate expression of those genes directly through recognizing and binding to DNA. Although PgRsp binds heme, no significant changes in PgRsp-DNA migration were observed when heme was added to the samples (Figure 9c).

Many proteins belonging to the Crp/Fnr superfamily autoregulate their own production [31], therefore we examined whether PgRsp protein can also regulate its own expression. For this purpose, the interaction of the purified PgRsp protein with the promoter region of the *pgrsp* gene was examined using EMSA analysis. Figure 9d showed that expression of the *pgrsp* gene could be autoregulated.

## 4. Discussion

The oral cavity is inhabited by microorganisms forming one of the most diverse microbiomes of the human body. Development of periodontitis is linked with environmental shift in the oral microbial consortium [58], leading to more reduced conditions and domination of Gram-negative bacteria over early colonizers, the latter mainly aerobic, Gram-positive bacteria (e.g., *S. gordonii*). *P. gingivalis*, as a secondary colonizer, has to interact with other bacteria and host cells to invade and colonize this niche and, importantly, to evade the host immune response. Therefore, precise regulation of gene expression is crucial for nutrient uptake and bacterial survival in a very hostile inflammatory environment within the oral microbiome and inside the host cells. Among a variety of changing environmental stimuli is redox status influenced by oxygen tension, nutrient availability (including iron and heme), and ROS and RNS levels. Oxidative stress is caused by metabolic processes occurring in bacteria and by agents present in the environment, and is generated by host cells, mostly neutrophils and macrophages, which produce ROS and H_2_O_2_ to eliminate bacteria, as well as by redox-active drugs used to combat pathogens. *P. gingivalis* cannot grow under aerobic conditions but is highly aerotolerant. To resist, replicate, and cause pathogenic processes, *P. gingivalis* has developed a variety of mechanisms to successfully cope with those harmful conditions.

For sensing and responding to the changes in the environment, pathogenic bacteria produce many transcription regulators [59]. They can act directly and/or indirectly on gene expression through the regulatory mechanism(s), often encompassing a multilayer regulatory network. In our previous studies we showed that PgFur regulates expression of virulence factors of *P. gingivalis* and is involved in invasion of human cells [13,16]. We proposed that PgFur is a part of a multilayer regulatory network, which also regulates expression of other transcription factors, including the gene encoding PgRsp protein [16]. Therefore, in this study we initiated characterization of PgRsp in regard to its involvement in potential regulatory mechanisms. Our results suggest that expression of the *pgrsp* gene is iron/heme- and growth phase-dependent. A similar gene expression pattern was observed for the *pgfur* gene [16]. Iron/heme homeostasis is closely linked to the oxidative stress response in *P. gingivalis* [60], which could explain the similar influence of external environmental conditions on the mechanism of action of PgFur and PgRsp proteins. Our findings showed that deletion of the *pgrsp* gene changes the interaction between *P. gingivalis* cells and other periodontopathogens. Recently, we reported similar phenotype for the *pgfur* mutant strain constructed in the A7436 strain [13,16]. We assume that these two proteins mutually regulate expression of genes involved in virulence in an iron/heme-dependent manner.

Besides changes in expression of genes selected in this study, we showed direct interaction of PgRsp protein with the promoter of the *hmu* operon, encoding six proteins, which are associated with heme uptake and its transport through the outer membrane [12,55]. It has been shown that HmuY protein is an important factor of *P. gingivalis* virulence [12,55,56]. It is expressed during bacterial invasion and proliferation inside the host cells [15]. Many studies have shown that HmuY protein is produced at higher levels in biofilm structures, thus enabling efficient uptake of heme [35,61,62]. PgRsp regulation of the Hmu heme acquisition system components and proteases involved in degradation of the host proteins suggests that it could be directly involved in regulation of virulence, allowing the bacterium to survive in the diverse microbiome of the oral cavity, as well as inside host cells. In addition, HmuY and hemagglutinin domains of Kgp and RgpA gingipains may provide more efficient heme accumulation on the bacterial cell surface, thus participating in formation of oxidative buffer to neutralize ROS [63,64]. Here we demonstrated that in the *pgrsp* mutant strain this ability was disturbed.

Sensing oxygen species and the redox state of the external environment is very important for anaerobic bacteria. One of the best known and characterized bacterial proteins involved in oxidative stress response and aerotolerance is OxyR [30,65,66,67]. It has been shown that also heme limitation may induce expression of the *oxyR* gene and genes regulated by this transcription factor, such as *sod*, *pgdps*, and *ahpC* [60,67]. Our study showed that also PgRsp is important for the *P. gingivalis* response to oxidative stress. We observed that heme limitation together with deletion of the *pgrsp* gene significantly altered expression of several genes involved in the oxidative stress response. The largest changes in the phenotype, as well as in gene expression, were observed when bacteria were grown in nutrient rich conditions and exposed to mild aerobic stress in the absence of cysteine.

It has been shown that Bcp and VimA proteins are important for anaerobic bacteria in regard to their resistance to air and H_2_O_2_ exposure; they are linked with gingipain expression [68], autoaggregation, and virulence [69]. Similar observations were made for the *pgrsp* mutant strain, demonstrating higher biofilm formation and coaggregation ability with other oral pathogens. Importantly, we also found that the *pgrsp*-deficient strain was less virulent in the macrophage infection model. Such regulation is possible since we observed direct interaction of PgRsp protein with the promoter of the *bcp* gene. All these data suggest that proteolytic activity together with heme accumulation on the bacterial cell surface could be coordinated to cope with harmful oxidative effects.

UstA protein is an important factor regulating gene expression during the stationary phase of bacterial growth. Interestingly, *P. gingivalis ustA* expression was higher in the stationary phase and under oxygen exposure [70]. It has been shown by others that deletion of the *ustA* gene resulted in impaired growth in rich culture medium and in changes in expression of genes involved in the response to oxidative stress [70]. In addition, exposure of the TO11 strain to mild oxidative conditions resulted in two times higher expression of the *ustA* gene, whereas H_2_O_2_ had an opposite impact on expression of the gene. Overexpression of the *ustA* gene could explain the growth of the TO11 strain, which produces significantly higher bacterial biomass compared to the A7436 strain. Based on results obtained in this study and observations presented by Abaibou and coworkers [68] and Johnson and coworkers [67] we assume that the operon encoding Bcp and VimA proteins, as well as the *ustA* gene are important targets for PgRsp regulatory activity.

Proteins from the Crp/Fnr superfamily play an important role in sensing stress conditions and are crucial for bacterial virulence [31,71]. To date, only one protein from this superfamily has been characterized in *P. gingivalis*. Lewis and coworkers [71] showed that HcpR protein (an Fnr homolog) regulates expression of genes responsible for the resistance of *P. gingivalis* to nitrosative stress by sensing NO through its interaction with heme bound to the HcpR protein [33]. We demonstrated that PgRsp protein belongs to the Crp/Fnr superfamily of transcription factors and could be classified as a relatively close homolog to the CooA family. Proteins homologous to CooA can sense CO [31,53], employing interaction of CO with heme bound to the protein [51,53], thus regulating genes important for utilization of CO as an energy source [72,73]. The molecular mechanism of the best-characterized homolog of CooA protein from *R. rubrum*, RrCooA, implies binding of Fe(III)heme by Pro2 and Cys75, when the protein is not active. Reduction of iron bound to heme causes replacement of coordinating Cys75 by His77. For protein activation, the CO molecule is bound to heme by a Pro2 residue, resulting in conformational changes of the protein, leading to interaction with DNA. This kind of protein activation is unique for RrCooA protein [51,52,53], since in the CooA homolog from *Carboxydothermus hydrogenoformans*, the heme molecule is coordinated by Ala2 and His82 residues and no transfer of the heme group occurs during the change of iron redox state [52,73]. 

Different redox state sensing mechanisms would be of great importance for bacteria for a fast response to changes in redox state of the external environment of the oral cavity or periodontal pockets, as well as inside the host cells. For example, PerR proteins sense hydrogen peroxide by metal-catalyzed His oxidation, which leads to conformational changes and dissociation from DNA and activation of expression of oxidative stress response genes [74,75]. On the other hand, in the classic OxyR paradigm, H_2_O_2_ oxidizes Cys residues, influencing the DNA–OxyR interaction, which leads to the activation of genes encoding proteins with antioxidant function. Similar feature of PgRsp could be important for *P. gingivalis* not only to resist oxidative stress generated by host immune cells during progression of periodontal diseases but importantly during metabolic disorders connected with chronic periodontitis [10]. However, our analysis of PgRsp point mutation variants affected in Cys76, Met78, or His85, did not allow us to confirm heme coordinating ligands since all protein variants examined bound heme with similar ability. Therefore we assume that the regulation mechanism employed by PgRsp differs compared to those used by other members of the Crp/Fnr superfamily of proteins.

The *P. gingivalis* response to the oxidizing environment is considered to be regulated mostly by OxyR protein. We assume that the mechanism of PgRsp protein response to the redox state could be important for sensing the redox state. It could imply activation by heme-iron-catalyzed oxidation similar to that observed in PerR protein or more likely by an activation process similar to that characterized for CooA protein with the exception that instead of an CO molecule, an O_2_ molecule could be bound to the heme iron. This would cause protein structure rearrangements and changes in the interaction of the PgRsp protein with DNA and/or PgFur. The proposed mechanism is also supported by studies demonstrating that some proteins from the CooA family are able to bind an oxygen molecule through heme and that they may regulate resistance to oxidative stress in anaerobic bacteria [34,75]. Based on our results, one may assume that PgRsp could be a redox-sensing protein, utilizing for this purpose heme bound to the protein. However, the heme-based mechanism of this regulation requires further investigation. Nevertheless, results gained in this study broaden knowledge on *P. gingivalis* gene regulation mechanisms, focused on its adaptation to different redox state environments.

## 5. Conclusions

PgRsp protein belongs to a novel family of heme-binding proteins, the Crp/Fnr superfamily of transcription regulators. Based on our data, one may conclude that this protein is a heme-based redox sensor which plays a role in the oxidative stress response and *P. gingivalis* virulence. However, at this stage of study we are unable to state whether PgRsp senses redox conditions (a more complex process) and/or oxygen (direct binding of an oxygen molecule). The amino acid identity and similarity of the PgRsp protein to other heme-binding proteins of the Crp/Fnr family is low; we assume that the heme-binding motif and transcriptional activation mechanism may differ from those known for CooA proteins. PgRsp as a redox-sensing protein could alter gene expression alongside or cooperate with OxyR protein, as well as with PgFur protein in the response and subsequent resistance to oxidative stress.

## Figures and Tables

**Figure 1 microorganisms-07-00623-f001:**
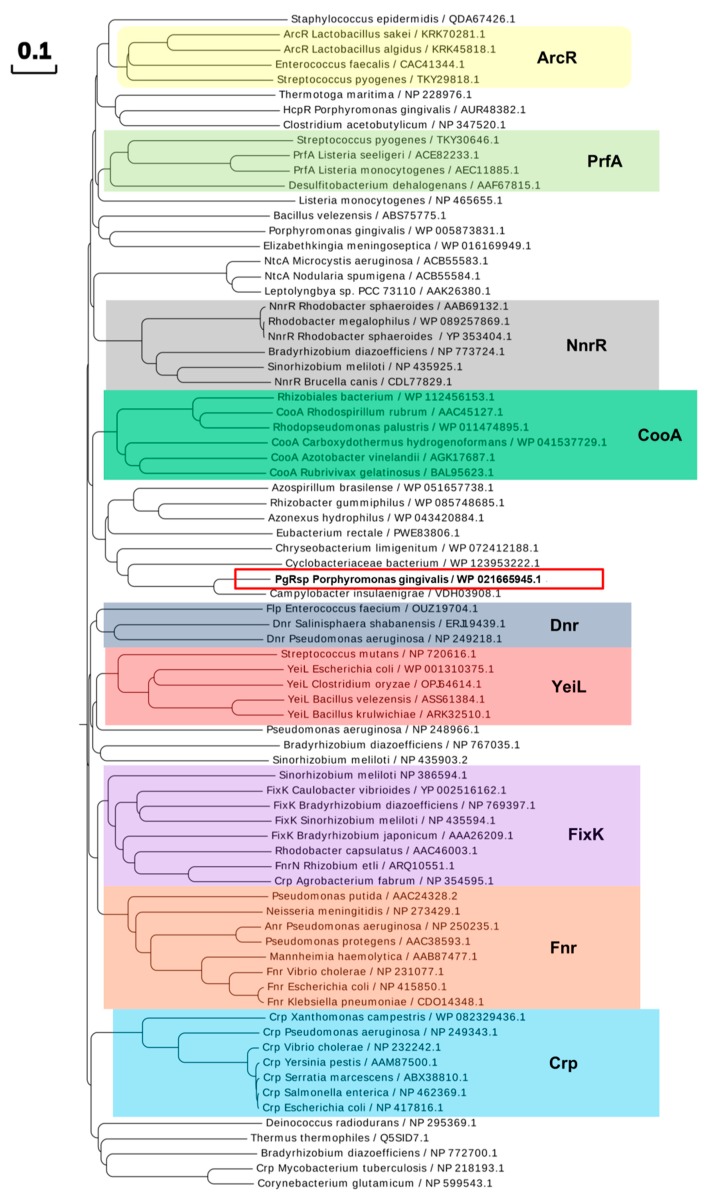
Phylogenetic analysis of the *P. gingivalis* PgRsp and selected proteins comprising Crp/Fnr superfamily. Representative proteins from chosen families are marked with colored shades. Homologous protein from *P. gingivalis* (PgRsp) is marked with a red frame. The proteins are shown by the bacterial species and NCBI accession number. Amino acid sequences were identified using BLAST search and compared using Clustal Omega and Simple Phylogens online tools. The phylogenetic tree was created using the online tool EvolView.

**Figure 2 microorganisms-07-00623-f002:**
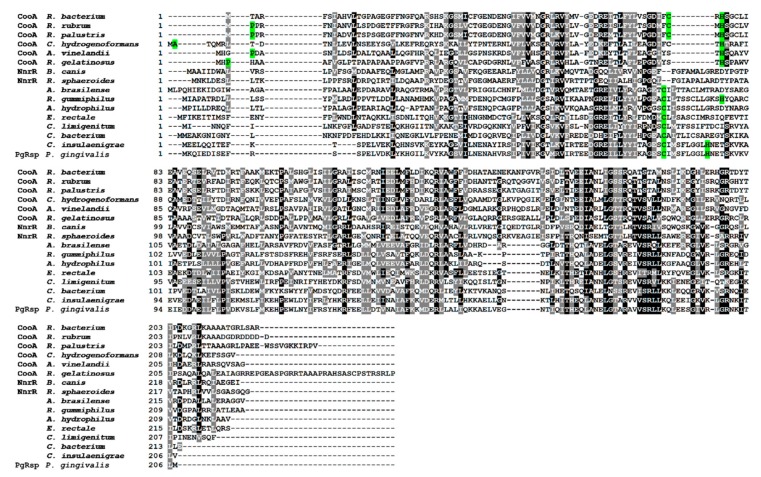
Comparison of amino acid sequences of the *P. gingivalis* PgRsp and selected proteins comprising Crp/Fnr superfamily. Amino acid residues involved in heme binding in *R. rubrum* CooA protein and putative heme-binding residues in other CooA homologs are marked in green. Black shadows indicate identical amino acids, gray shadows indicate amino acids with similar properties. Amino acid sequences were identified using BLAST search and compared using the Clustal Omega and the BoxShade server.

**Figure 3 microorganisms-07-00623-f003:**
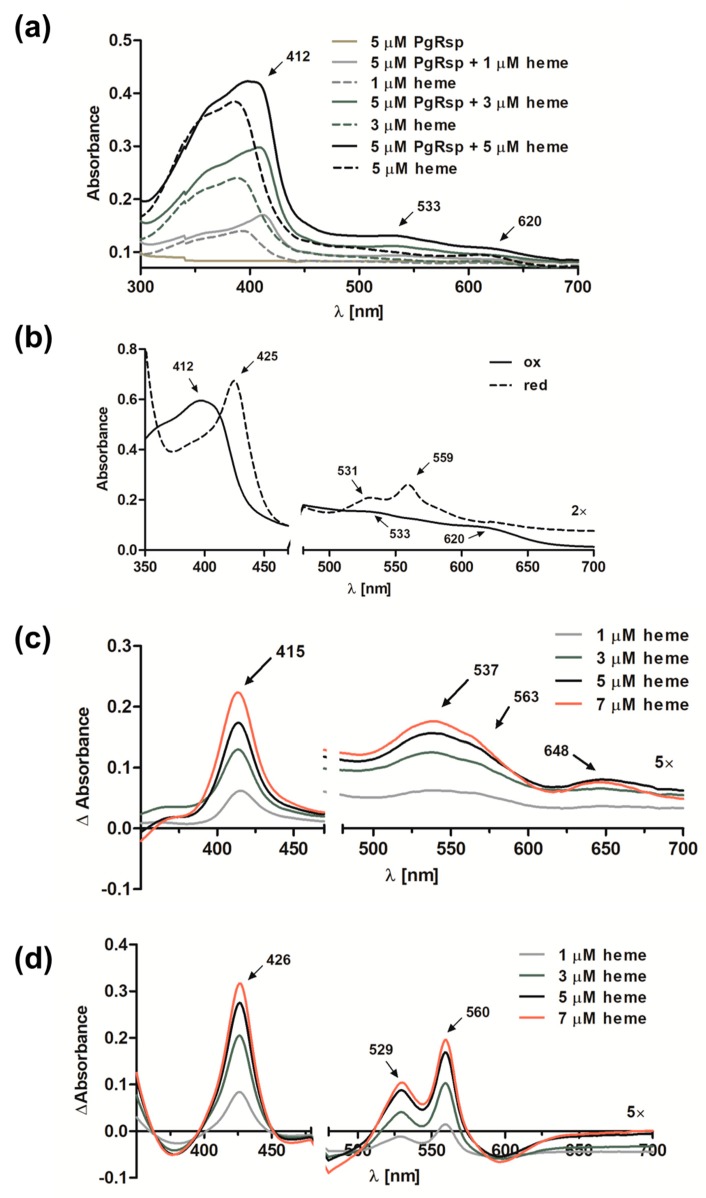
Heme binding to the purified PgRsp protein. (**a**) UV-visible absorbance spectra recorded after titration of 5 µM PgRsp with heme under air (oxidized conditions). (**b**) Spectra recorded for 10 µM PgRsp complexed with heme (1:0.8 protein:heme ratio) under oxidized and reduced conditions. (**c**,**d**) Difference spectra after titration of 5 µM PgRsp with heme under (**c**) oxidized and (**d**) reduced conditions. All experiments were performed three times and representative spectra are shown.

**Figure 4 microorganisms-07-00623-f004:**
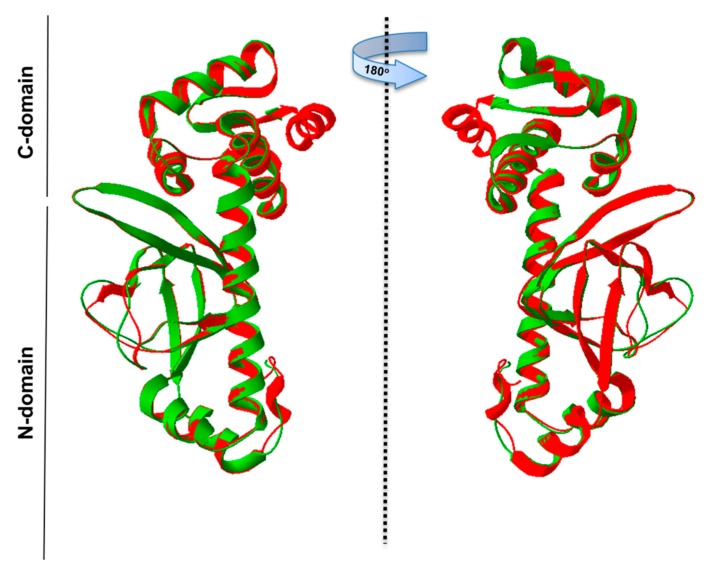
Theoretical three-dimensional model of PgRsp protein structure. PgRsp shows fold typical for Crp/Fnr homologs: it contains a C-terminal DNA-binding domain connected with an N-terminal, sensor domain, and alpha-helix connecting the two domains, which is responsible for protein dimerization. The structure was modeled using the Phyre2 web-based modeling server based on the three-dimensional protein structure (PDB ID: 3DV8) of the *Eubacterium rectale* Crp/Fnr superfamily member as a template. Green—PgRsp theoretical model; red—experimentally solved structure of *E. rectale* protein.

**Figure 5 microorganisms-07-00623-f005:**
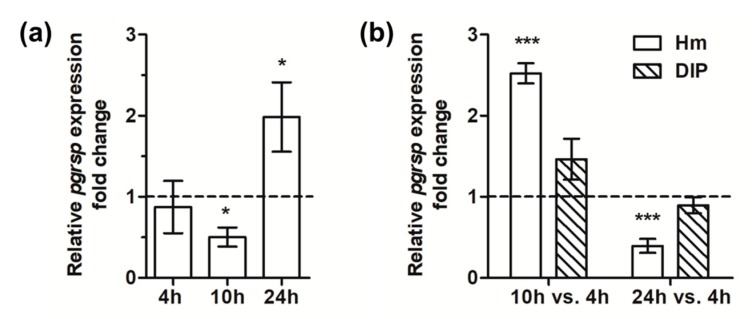
Analysis of *pgrsp* gene expression. Relative expression fold change in the transcript level was determined by RT-qPCR in *P. gingivalis* wild-type A7436 strain in relation to iron/heme presence (bacteria cultured in BM + DIP vs. bacteria grown in BM + Hm) (**a**) or growth phase (**b**). Bacteria were cultured in BM + Cys medium supplemented with 7.7 µM hemin (Hm) or lacking hemin and supplemented with 160 µM dipyridyl (DIP). Experiments were performed three times in two biological replicates. Representative results are shown as mean ± standard deviation. The dotted line shows the mean expression of the gene under control conditions. Statistically significant results were indicated with asterisks (* *p* < 0.05; *** *p* < 0.001).

**Figure 6 microorganisms-07-00623-f006:**
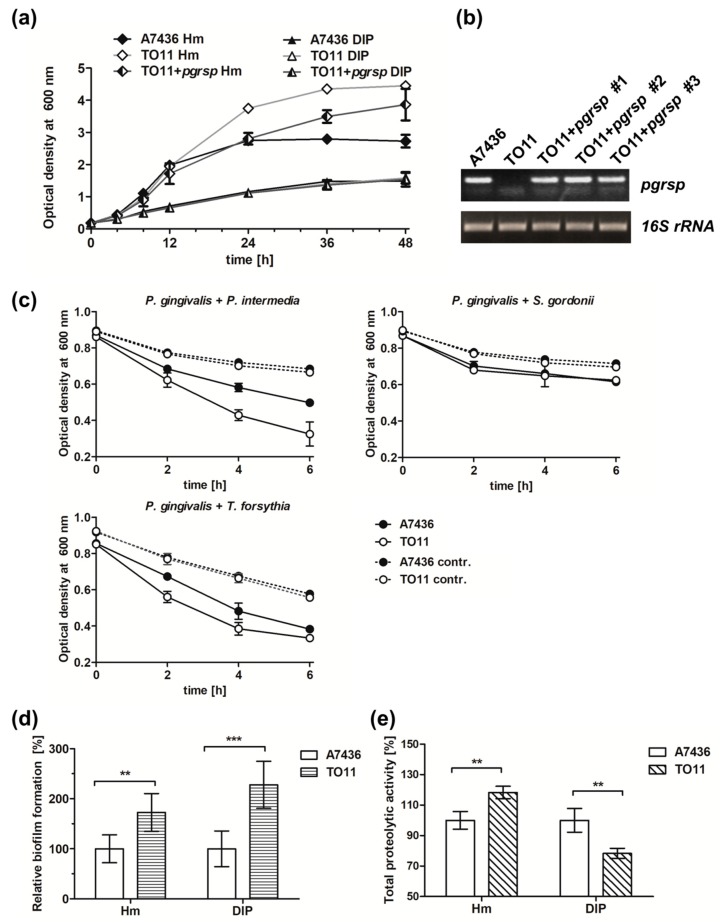
Phenotypic characterization of *P. gingivalis* wild-type (A7436), *pgrsp* mutant (TO11), and complemented mutant (TO11 + *pgrsp*) strains. All strains were grown in BM + Cys medium, supplemented with 7.7 µM hemin (Hm) or lacking hemin and supplemented with 160 µM dipyridyl (DIP) for 48 h. (**a**) Bacterial growth was monitored by measuring OD at 600 nm. (**b**) *pgrsp* transcript level was determined using RT-PCR and compared with *16S rRNA* gene expression. (**c**) *P. gingivalis* coaggregation with *P. intermedia*, *T. forsythia,* or *S. gordonii* was monitored by measuring the decrease in OD at 600 nm. Autoaggregation of monocultures was used as a control. (**d**) Relative biofilm formation ability on the abiotic surfaces and (**e**) total proteolytic activity were measured in 24-h bacterial cultures in iron/heme-rich (Hm) or iron/heme-depleted conditions (DIP). All experiments were performed three times in at least two biological replicates. Results are shown as mean ± standard deviation. Statistically significant results were indicated with asterisks (** *p* < 0.01; *** *p* < 0.001).

**Figure 7 microorganisms-07-00623-f007:**
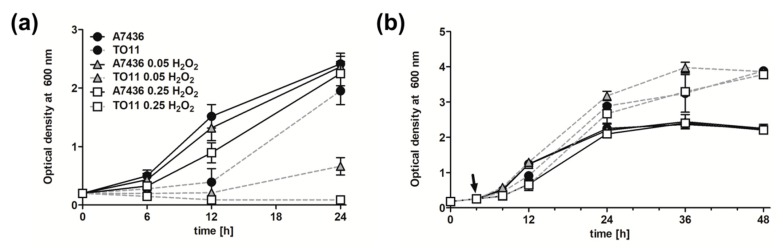
Growth characteristics of *P. gingivalis pgrsp* mutant strain cultured under different stress conditions. Wild-type (A7436) and *pgrsp* mutant (TO11) strains were grown in BM medium supplemented with hemin but lacking cysteine (BM–Cys + Hm), with or without 0.05 mM or 0.25 mM H_2_O_2_ added (**a**) at the starting culture point or (**b**) after 4 h (indicated by arrow). Bacterial growth was monitored by measuring OD at 600 nm. All experiments were performed three times in at least two biological replicates. Representative results are shown as mean ± standard deviation.

**Figure 8 microorganisms-07-00623-f008:**
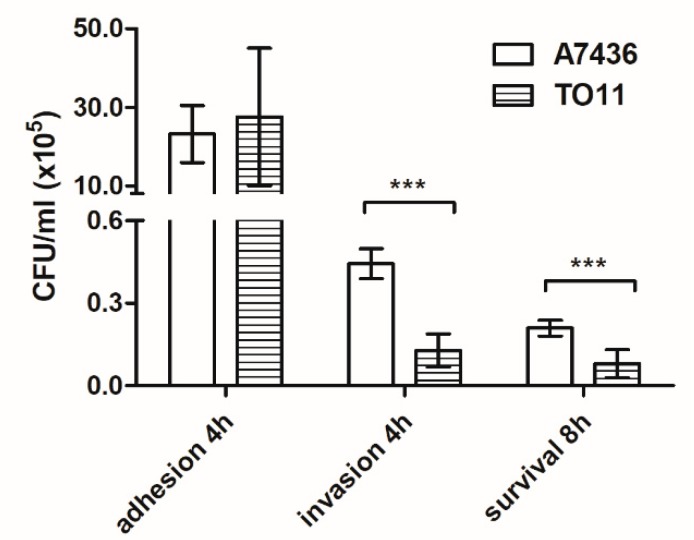
Comparison of the ability of *P. gingivalis* wild-type (A7436) and *pgrsp* mutant (TO11) strains to interact with, invade, and survive inside THP-1 cells differentiated towards macrophages. The number of viable bacteria was determined per milliliter of cell culture at the appropriate time points. Adhesion—the number of live bacteria attached to the cell surface after 4 h; invasion—the number of live bacteria that invaded the cells after 4 h; survival—the number of live bacteria inside the cells after 8 h. Experiments were performed three times and representative results are shown as mean ± standard deviation. Statistically significant results were indicated with asterisks (*** *p* < 0.001).

**Figure 9 microorganisms-07-00623-f009:**
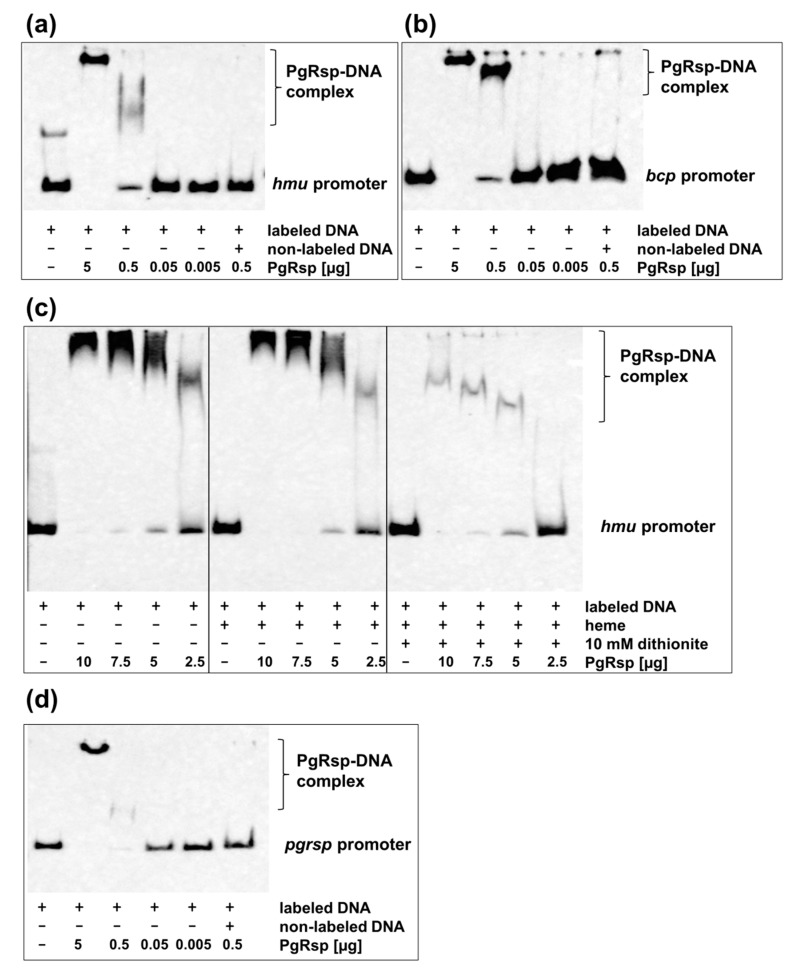
Direct regulation of genes by PgRsp protein. Binding of the purified PgRsp protein to the (**a**,**c**) *hmu*, (**b**) *bcp*, and (**d**) *pgrsp* promoters was examined using electrophoretic mobility shift assay (EMSA). (**c**) Influence of heme and different redox condition on PgRsp interaction with operon *hmu* promoter. Labeled DNA (1 ng) was incubated with varying amounts of the purified PgRsp protein in the presence of 10 µM heme, 10 µM cAMP, and 10 mM sodium dithionite. The control sample contained addition of 50-fold excess of non-labeled examined promoter DNA. Experiments were performed three times and representative results are shown.

**Table 1 microorganisms-07-00623-t001:** Analysis of gene expression in *P. gingivalis pgrsp* mutant strain (TO11) in comparison to the wild-type A7436 strain during bacterial growth under iron/heme-rich (Hm) or iron/heme-depleted (DIP) conditions. Data are marked as mean ± standard deviation. Statistically significant changes are marked with an asterisk.

Gene	Hm			DIP		
	4 h	10 h	24 h	4 h	10 h	24 h
*hmuY*	−1.41 ± 0.24	1.51 ± 0.33 *	1.35 ± 0.35	1.04 ± 0.05	1.28 ± 0.27	−1.06 ± 0.16
*rgpB*	−1.52 ± 0.10 *	−1.16 ± 0.23	1.75 ± 0.35 *	−1.80 ± 0.05 *	1.12 ± 0.24	−1.05 ± 0.22
*kgp*	1.62 ± 0.14 *	1.80 ± 0.35 *	4.10 ± 0.77 *	1.14 ± 0.11	2.48 ± 0.52 *	2.77 ± 0.33 *
*rgpA*	−1.53 ± 0.10 *	−1.57 ± 0.32 *	1.50 ± 0.27 *	−2.83 ± 0.17 *	−1.51 ± 0.32 *	−1.47 ± 0.22
*PGA7_00013810*	−1.33 ± 0.11	−1.61 ± 0.30 *	−1.87 ± 0.26 *	−2.22 ± 0.52 *	−1.48 ± 0.36	−1.09 ± 0.14
*fimA*	−1.57 ± 0.21	−1.03 ± 0.25	−2.22 ± 0.84 *	1.72 ± 0.09 *	−1.34 ± 0.19	−1.69 ± 0.32 *
*sod*	−1.09 ± 0.17	3.04 ± 0.52 *	1.56 ± 0.57 *	−1.50 ± 0.06	2.11 ± 0.33 *	1.02 ± 0.20
*bcp*	1.60 ± 0.60 *	1.13 ± 2.00	1.59 ± 0.13 *	2.29 ± 0.12 *	1.95 ± 0.22 *	2.17 ± 0.11 *
*pgdps*	−1.07 ± 0.18	−1.29 ± 3.00	1.38 ± 0.40	1.49 ± 0.60	1.22 ± 0.70	1.15 ± 0.10
*ahpC*	−1.14 ± 0.50	−1.30 ± 0.13	1.80 ± 0.40 *	1.05 ± 0.21	2.66 ± 0.90 *	1.05 ± 0.12
*rbr*	1.47 ± 0.11	1.56 ± 0.15	1.31 ± 0.60	2.18 ± 0.16 *	1.61 ± 0.14	1.93 ± 0.23 *
*tpx*	1.69 ± 0.13 *	1.31 ± 0.40	2.62 ± 0.50 *	4.88 ± 0.27 *	3.90 ± 0.29 *	1.81 ± 0.10 *
*oxyR*	1.64 ± 0.23 *	1.11 ± 0.13	−1.05 ± 0.42	1.08 ± 0.10	1.64 ± 0.14 *	1.56 ± 0.15 *
*pgfur*	1.14 ± 0.05	1.18 ± 0.16	1.12 ± 0.25	−1.05 ± 0.24	1.33 ± 0.33	1.60 ± 0.15 *
*ustA*	3.52 ± 2.49	−1.23 ± 0.59	2.69 ± 0.82*	1.11 ± 0.41	4.52 ± 1.27*	2.88 ± 2.24*

**Table 2 microorganisms-07-00623-t002:** Analysis of gene expression in response to different redox conditions during growth in liquid culture medium of the *P. gingivalis pgrsp* mutant strain (TO11) in comparison to the wild-type A7436 strain. Data are marked as mean ± standard deviation. Changes statistically significant are marked with an asterisk.

Gene	BM + Cys + Hm	BM−Cys + Hm	BM−Cys + Hm + 0.25 mM H_2_O_2_
*hmuY*	−1.76 ± 0.59 *	−14.03 ± 12.69 *	4.44 ± 3.41 *
*rgpB*	1.28 ± 0.52	−4.16 ± 2.25 *	5.62 ± 3.61 *
*kgp*	−1.09 ± 0.31	−4.29 ± 2.71 *	1.40 ± 1.64
*rgpA*	−1.47 ± 0.17	−3.05 ± 0.88 *	−1.20 ± 0.32
*PGA7_00013810*	1.03 ± 0.35	−4.40 ± 2.64 *	−3.36 ± 2.50 *
*fimA*	−1.54 ± 0.34	−3.00 ± 2.98 *	−3.44 ± 1.82 *
*sod*	2.74 ± 0.62 *	−15.72 ± 14.21 *	1.04 ± 0.84
*bcp*	1.21 ± 0.12	−2.86 ± 1.58 *	−5.49 ± 3.71 *
*pgdps*	1.01 ± 0.80	1.29 ± 0.32	−4.38 ± 2.78 *
*ahpC*	1.73 ± 0.19 *	−1.53 ± 0.44 *	−1.70 ± 0.96
*rbr*	−1.07 ± 0.08	−1.27 ± 0.15	2.60 ± 1.61 *
*tpx*	2.72 ± 0.79 *	−5.40 ± 4.91 *	1.99 ± 0.85 *
*oxyR*	−1.16 ± 0.21	−1.52 ± 0.49	7.39 ± 6.10 *
*pgfur*	−1.20 ± 0.24	−3.24 ± 1.69 *	1.07 ± 0.57
*ustA*	−1.22 ± 0.49	2.16 ± 0.90 *	−1.83 ± 0.70 *

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
