# Peer review of "PgRsp Is a Novel Redox-Sensing Transcription Regulator Essential for Porphyromonas gingivalis Virulence"

_microorganisms, 2019, doi:10.3390/microorganisms7120623_

Round 1

Reviewer 1 Report

Dear authors,

I read your manuscript with interest and I think it is interesting and innovative.

Beside this I found some conceptual improprieties that should be addressed before publication.

here you can find them in details:

Abstract: Line 10: "Porphyromonas gingivalis is considered the main etiological agent of chronic periodontitis." Please avoid such type of sentences as it is clearly demonstrated that Porphyromonas gingivalis is merely one of the etiological agents of chronic periodontitis (see subsequent comments for further considerations); Introduction: Line 31: "reduction of tooth-supporting tissues caused by gum inflammation". Periodontal disease pathogenesis deals with the "reabsorption" of tooth-supporting tissues (alveolar bone, root cementum and periodontal ligament); writing about "reduction" is a lexical impropriety especially if you refer to gum (colloquial term that should be substituted by "gingiva") that it is forced to migrate apically, not to reduct; Introduction: Line 34-35: "as cardiovascular and respiratory diseases, rheumatoid arthritis, diabetes, osteoporosis, and Alzheimer’s disease". Please add recent evidence regarding connections between metabolic syndrome and periodontal disease (for your convenience: Patini R et al.: Correlation between metabolic syndrome, periodontitis and reactive oxygen species production. A pilot study. Open Dent J. 2017;11:621-627); Introduction: Line 35-36: "Porphyromonas gingivalis, considered the main etiologic agent of chronic periodontitis". Same considerations as in abstract: This statement is not completely correct (or in any case obsolete) in fact it is supported by evidences of 2011-2012. According to the most recent evidence chronic periodontitis is supported by a complex of non-specific oral microflora. Among bacteria that compose such microflora it has been recently clearly demonstrated that Porphyromonas Gingivalis is present but its relative concentration seems to be surpassed by Tannerella Forsythensis and Filifactor Alocis. Please cite appropriate references (for your convenience: Patini R et al.: Relationship between oral microbiota and periodontal disease: A systematic review. Eur Rev Med Pharmacol Sci. 2018;22(18):5775-5788); Introduction: Line 38-40: "It is a component of the multispecies oral microbiome, also containing other members of the “red complex” (Tannerella forsythia  and Treponema denticola), as well as representatives of the “orange complex” (e.g., Prevotella intermedia) and “yellow complex” (e.g., Streptococcus gordonii  and S. oralis)". Such information dates back many years ago and can no longer be considered current. The classification into "complexes" has only a mere historical value; I suggest to delete this sentence; Introduction: I suggest to shorten and to add the final paragraph in which the final aim of the work must be reported. Without reporting it clearly the global comprehension of the manuscript is really affected. Introduction: Line 87-89: "We found that the phenotype of the bacteria lacking this protein is significantly altered when exposed to oxidative stress. Therefore, we termed this protein a P. gingivalis  redox sensing  protein (PgRsp)." Please move these sentences to results sections; Introduction: Line 89-90: "Results gained in this study broaden knowledge on P. gingivalis  gene regulation mechanisms, focused on its adaptation to different redox state environments." Please move this sentence to discussion section; Materials and Methods: Line 93: "P. gingivalis wild type (A7436)" please give ATCC code for P. gingivalis; Materials and Methods: Line 104: please give ATCC code for P. intermedia; Materials and Methods: Line 254: please avoid using personal pronouns (e.g. we) in favor of a more impersonal way of writing (suggestion: it has been compared); Discussion: Line 516: "PgRsp protein [16 and this study]". You cannot cite your own work. Please delete. Discussion: In the discussion (which must be reduced because it cannot only be an examination of the existing literature on the subject but an interpretation of the results of the study in light of what is known in the literature) you have to report the hypothetical clinical repercussion that this experimentation could have (also known as "generalizability" and "external validity").

I hope you can fix all my concerns.

Best regards

Author Response

Point 1: Abstract: Line 10: "Porphyromonas gingivalis is considered the main etiological agent of chronic periodontitis." Please avoid such type of sentences as it is clearly demonstrated that Porphyromonas gingivalis is merely one of the etiological agents of chronic periodontitis (see subsequent comments for further considerations);

Response 1: The sentence was changed and marked in the manuscript.

Point 2: Introduction: Line 31: "reduction of tooth-supporting tissues caused by gum inflammation". Periodontal disease pathogenesis deals with the "reabsorption" of tooth-supporting tissues (alveolar bone, root cementum and periodontal ligament); writing about "reduction" is a lexical impropriety especially if you refer to gum (colloquial term that should be substituted by "gingiva") that it is forced to migrate apically, not to reduct;

Response 2: The reviewer suggestion was taken into consideration and appropriate changes were made and marked in the manuscript.

Point 3: Introduction: Line 34-35: "as cardiovascular and respiratory diseases, rheumatoid arthritis, diabetes, osteoporosis, and Alzheimer’s disease". Please add recent evidence regarding connections between metabolic syndrome and periodontal disease (for your convenience: Patini R et al.: Correlation between metabolic syndrome, periodontitis and reactive oxygen species production. A pilot study. Open Dent J. 2017;11:621-627);

Response 3: The suggested citation was added.

Point 4: Introduction: Line 35-36: "Porphyromonas gingivalis, considered the main etiologic agent of chronic periodontitis". Same considerations as in abstract: This statement is not completely correct (or in any case obsolete) in fact it is supported by evidences of 2011-2012. According to the most recent evidence chronic periodontitis is supported by a complex of non-specific oral microflora. Among bacteria that compose such microflora it has been recently clearly demonstrated that Porphyromonas Gingivalis is present but its relative concentration seems to be surpassed by Tannerella Forsythensis and Filifactor Alocis. Please cite appropriate references (for your convenience: Patini R et al.: Relationship between oral microbiota and periodontal disease: A systematic review. Eur Rev Med Pharmacol Sci. 2018;22(18):5775-5788);

Response 4: The sentence was changed and marked in the manuscript and the suggested citation was added.

Point 5: Introduction: Line 38-40: "It is a component of the multispecies oral microbiome, also containing other members of the “red complex” (Tannerella forsythia  and Treponema denticola), as well as representatives of the “orange complex” (e.g., Prevotella intermedia) and “yellow complex” (e.g., Streptococcus gordonii  and S. oralis)". Such information dates back many years ago and can no longer be considered current. The classification into "complexes" has only a mere historical value; I suggest to delete this sentence;

Response 5: The sentence was deleted.

Point 6: Introduction: I suggest to shorten and to add the final paragraph in which the final aim of the work must be reported. Without reporting it clearly the global comprehension of the manuscript is really affected.

Response 6: The changes were made and marked in the manuscript.

Point 7: Introduction: Line 87-89: "We found that the phenotype of the bacteria lacking this protein is significantly altered when exposed to oxidative stress. Therefore, we termed this protein a P. gingivalis  redox sensing  protein (PgRsp)." Please move these sentences to results sections;

Response 7: The sentences were moved as requested.

Point 8: Introduction: Line 89-90: "Results gained in this study broaden knowledge on P. gingivalis  gene regulation mechanisms, focused on its adaptation to different redox state environments." Please move this sentence to discussion section;

Response 8: The sentences were moved as requested.

Point 9: Materials and Methods: Line 93: "P. gingivalis wild type (A7436)" please give ATCC code for P. gingivalis;

Response 9: The A7436 is a routinely used strain, isolated from an individual affected with periodontitis. The citation for the study is listed below. For clarity, the citation was included in the manuscript.

Genco, C.A.; Cutler, C.W.; Kapczynski, D.; Maloney, K.; Arnold, R.R. A novel mouse model to study the virulence of and host response to Porphyromonas (Bacteroides) gingivalis. Infect Immun. 1991 59(4):1255-63

Point 10: Materials and Methods: Line 104: please give ATCC code for P. intermedia;

Response 10: The P. intermedia strain 17 is a clinical strain isolated from human periodontal pockets. For clarity, the accesion numbers (GenBank: CP003502.1 and CP003503.1) were added to the manuscript.

Point 11: Materials and Methods: Line 254: please avoid using personal pronouns (e.g. we) in favor of a more impersonal way of writing (suggestion: it has been compared);

Response 11: The suggestion was taken into consideration and changes are marked in the manuscript.

Point 12: Discussion: Line 516: "PgRsp protein [16 and this study]". You cannot cite your own work. Please delete.

Response 12: The citation was removed.

Point 13: Discussion: In the discussion (which must be reduced because it cannot only be an examination of the existing literature on the subject but an interpretation of the results of the study in light of what is known in the literature) you have to report the hypothetical clinical repercussion that this experimentation could have (also known as "generalizability" and "external validity").

Response 13: When possible, the Discussion section was shorten. We also added reference and short discussion on this subject.

General comment: the manuscript was corrected previously by the native speaker, familiar with biological sciences. After changes made in the revised text, the manuscript was double checked.

Reviewer 2 Report

I have reviewed an article: "PgRsp is a novel redox sensing transcription regulator essential for Porphyromonas gingivalis virulence".

It is very interesting and novel research that is based on complex experimental design that nicely mimics possible events in oral cavity. Results are nicely presented, Discussion is supported by experimental data and well compared by facts that are discovered by other scientists.Conclusions are based on experimental data and I think this paper should be accepted for publication in present form.

Author Response

I have reviewed an article: "PgRsp is a novel redox sensing transcription regulator essential for Porphyromonas gingivalis virulence".

It is very interesting and novel research that is based on complex experimental design that nicely mimics possible events in oral cavity. Results are nicely presented, Discussion is supported by experimental data and well compared by facts that are discovered by other scientists.Conclusions are based on experimental data and I think this paper should be accepted for publication in present form.

We would like to thank the reviewer for the positive reception of our manuscript.

General comment: the manuscript was corrected previously by the native speaker, familiar with biological sciences. After changes made in the revised text, the manuscript was double checked.

Round 2

Reviewer 1 Report

Dear authors,

  thanks for having satisfactorily addressed all my concerns.

I noticed that in the final paragraph of the introduction you added you have included some sentences regarding the results of the study. I remind you that in this section you must not report absolutely any data concerning the results but only specify the objective of your study and the null hypothesis.
Move the sentences on the results to the appropriate section or delete them.

Regards